# FAST STOCHASTIC KERNEL APPROXIMATION BY DUAL WASSERSTEIN DISTANCE METHOD

## ABSTRACT

We introduce a generalization of the Wasserstein metric, originally designed for probability measures, to establish a novel distance between probability kernels of Markov systems. We illustrate how this kernel metric may serve as the foundation for an efficient approximation technique, enabling the replacement of the original system's kernel with a kernel with a discrete support of limited cardinality. To facilitate practical implementation, we present a specialized dual algorithm capable of constructing these approximate kernels quickly and efficiently, without requiring computationally expensive matrix operations. Finally, we demonstrate the effectiveness of our method through several illustrative examples, showcasing its utility in diverse practical scenarios, including dynamic risk estimation. This advancement offers new possibilities for the streamlined analysis and manipulation of Markov systems represented by kernels.

## 1 INTRODUCTION

Consider a discrete-time Markov system described by the relations:

$$X_{t+1} \sim Q_t(X_t), \quad t = 0, 1, \ldots, T-1, \tag{1}$$

where $X_t \in \mathscr{X}$ is the state at time $t$, and $Q_t : \mathscr{X} \to \mathscr{P}(\mathscr{X})$, $t = 0, 1, \ldots, T-1$, are stochastic kernels. The symbol $\mathscr{X}$ represents a separable metric space (the state space), and $\mathscr{P}(\mathscr{X})$ is the space of probability measures on $\mathscr{X}$. Formula (1) means that the conditional distribution of $X_{t+1}$, given $X_t = x$, is $Q_t(x)$. The distribution of the initial state $\delta_{x_0}$ (the Dirac delta at $x_0$) and the sequence of kernels $Q_t$, $t = 0, \ldots, T-1$, define a probability measure $P$ on the space of paths $\mathscr{X}^{T+1}$.

One of the challenges of dealing with models of the form (1) is the need to evaluate a backward system (with a sequence of functions $c_t : \mathscr{X} \to \mathbb{R}$):

$$v_t(x) = c_t(x) + \sigma_t\big(x, Q_t(x), v_{t+1}(\cdot)\big), \quad x \in \mathscr{X}, \quad t = 0, \ldots, T-1;$$
$$v_T(x) = c_T(x), \quad x \in \mathscr{X}. \tag{2}$$

Problems of this type arise in manifold applications, such as financial option pricing, risk evaluation, and other dynamic programming problems. They are particularly difficult when the operators $\sigma_t(\cdot, \cdot, \cdot)$ are nonlinear with respect to the probability measures $Q_t(x)$ involved.

In equation (2), the operator $\sigma_t : \mathscr{X} \times \mathscr{P}(\mathscr{X}) \times \mathscr{V} \to \mathbb{R}$, where $\mathscr{V}$ is a space of Borel measurable real functions on $\mathscr{X}$, is a *transition risk mapping*. Its first argument is the present state $x$. The second argument is the probability distribution $Q_t(x)$ of the state following $x$ in the system (1). The last argument, the function $v_{t+1}(\cdot)$, is the next state's value: the risk of running the system from the next state in the time interval from $t+1$ to $T$.

A simple case of the transition risk mapping is the bilinear form,

$$\sigma_t\big(x, \mu, v_{t+1}(\cdot)\big) = \mathbb{E}_\mu\big[v_{t+1}(\cdot)\big]. \tag{3}$$

In this case, the scheme (2) evaluates the conditional expectation of the total cost from stage $t$ to the end of the horizon $T$:

$$v_t(x) = \mathbb{E}\big[c_t(X_t) + \cdots + c_T(X_T) \big| X_t = x\big], \quad x \in \mathscr{X}, \quad t = 0, \ldots, T.$$

A more interesting application is the *optimal stopping problem*, in which $c_t(\cdot) \equiv 0$, and

$$\sigma_t\big(x, \mu, v_{t+1}(\cdot)\big) = \max\Big(r_t(x)\,;\, \mathbb{E}_\mu\big[v_{t+1}(\cdot)\big]\Big). \tag{4}$$

Here, $r_t : \mathscr{X} \to \mathbb{R}$, $t = 0, \dots, T$, represent the rewards collected if the decision to stop at time $t$ and state $x$ is made. Clearly, with the mappings (4) used in the scheme (2),

$$v_t(x) = \sup_{\substack{\tau - \text{stopping time} \\ t \le \tau \le T}} r_\tau(X_\tau), \quad x \in \mathscr{X}, \quad t = 0, \dots, T;$$

see, *e.g.*, Chow et al. (1971). The most important difference between (3) and (4) is that the latter is nonlinear with respect to the probability measure $\mu$.

One of the challenges associated with the backward system (2) is the numerical solution in the case when the transition risk mappings are nonlinear with respect to the probability measures involved. The objective of this paper is to present a computational method based on approximating the kernels $Q_t(\cdot)$ by simpler, easier-to-handle kernels $\widetilde{Q}_t(\cdot)$, and using them in the backward system (2).

The approximation of stochastic processes in discrete time has attracted the attention of researchers for many decades. Fundamental in this respect is the concept of a *scenario tree*. Høyland & Wallace (2001) uses statistical parameters, such as moments and correlations, to construct such a tree. Kaut & Wallace (2011) involve copulas to capture the shape of the distributions. Heitsch & Römisch (2009) were probably the first to use probability metrics for reducing large scenario trees. Pflug (2010) introduced the concept of nested distance, using an extension of the Wasserstein metric for processes; see also (Pflug & Pichler, 2015). All these approaches differ from our construction in the Markovian case.

The Wasserstein distance has shown promising results in various applications such as Generative Adversarial Networks (GAN) (Arjovsky et al., 2017), clustering (Ho et al., 2017), semi-supervised learning (Solomon et al., 2014), and image retrievals (Rubner et al., 2000; Pele & Werman, 2009), among others. Some recent contributions measure the distance of mixture distributions rather than kernels. Bing et al. (2022) propose the sketched Wasserstein distance, a type of distance metric dedicated to finite mixture models. Research on Wasserstein-based distances tailored to Gaussian mixture models is reported in (Chen et al., 2020; Delon & Desolneux, 2020; Kolouri et al., 2018).

In parallel, we see continuous efforts to develop fast algorithms for computing the relevant transportation distances. One notable contribution is the Sinkhorn algorithm, introduced by Cuturi (2013), which incorporates an entropic regularization term to the mass transportation problem. Since then, both the Sinkhorn algorithm and its variant Greenkhorn (Altschuler et al., 2017) have become the baseline approaches for computing transportation distance and have triggered significant progress (Genevay et al., 2016; Lin et al., 2019). Other relevant approaches include accelerated primal-dual gradient descent (APDAGD) (Dvurechensky et al., 2018; Dvurechenskii et al., 2018; Kroshnin et al., 2019) and semi-dual gradient descent (Cuturi & Peyré, 2018; 2016).

**Contribution.**    This paper makes a threefold contribution. First, we introduce a kernel distance based on the Wasserstein distance between distributions. Second, we propose a new particle selection method that recursively approximates the forward system using the kernel distance. Third, we propose a decomposable, parallelizable subgradient algorithm for particle selection, avoiding constraints and matrix computations. We conduct extensive experiments and show that the subgradient algorithm performs favorably in practice.

**Organization.**    In section 2, we provide a brief overview of the distance metrics, including the Wasserstein and kernel distances. We also introduce the problem of selecting representative particles using a mixed-integer formulation based on distance metrics. In section 3, we present our subgradient method, its relation to the dual problem, and the algorithm used for selecting particles. In section 4, we present a numerical example. We use particles generated by the subgradient method to solve an optimal stopping problem based on a multi-dimensional stochastic process. In section 5, we conclude this paper.

## 2 THE PROBLEM

### 2.1 WASSERSTEIN DISTANCE

Let $d(\cdot,\cdot)$ be the metric on $\mathscr{X}$. For two probability measures $\mu,\nu$ on $\mathscr{X}$ having finite moments up to order $p \in [1,\infty)$, their Wasserstein distance of order $p$ is defined by the following formula (see (Rachev & Rüschendorf, 1998; Villani, 2009) for a detailed exposition and historical account):

$$W_p(\mu,\nu) = \left( \inf_{\pi \in \Pi(\mu,\nu)} \int_{\mathscr{X} \times \mathscr{X}} d(x,y)^p \, \pi(\mathrm{d}x,\mathrm{d}y) \right)^{1/p}, \tag{5}$$

where $\Pi(\mu,\nu)$ is the set of all probability measures in $\mathscr{P}(\mathscr{X} \times \mathscr{X})$ with the marginals $\mu$ and $\nu$.

We restrict the space of probability measures to measures with finite moments up to order $p$. Formally, we define the Wasserstein space:

$$\mathscr{P}_p(\mathscr{X}) := \left\{ \mu \in \mathscr{P}(\mathscr{X}) : \int_{\mathscr{X}} d(x_0,x)^p \, \mu(dx) < +\infty \right\}.$$

For each $p \in [1,\infty)$, the function $W_p(\cdot,\cdot)$ defines a metric on $\mathscr{P}_p(\mathscr{X})$. Furthermore, for all $\mu,\nu \in \mathscr{P}_p(\mathscr{X})$ the optimal coupling realizing the infimum in (5) exists. From now on, $\mathscr{P}_p(\mathscr{X})$ will be always equipped with the distance $W_p(\cdot,\cdot)$.

For discrete measures, problem (5) has a linear programming representation. Let $\mu$ and $\nu$ be supported at positions $\{x^{(i)}\}_{i=1}^N$ and $\{z^{(k)}\}_{k=1}^M$, respectively, with normalized (totaling 1) positive weight vectors $w_x$ and $w_z$: $\mu = \sum_{i=1}^N w_x^{(i)} \delta_{x^{(i)}}$, $\nu = \sum_{k=1}^M w_z^{(k)} \delta_{z^{(k)}}$. For $p \geq 1$, let $D \in R_+^{N \times M}$ be the distance matrix with elements $d_{ik} = d\big(x^{(i)}, z^{(k)}\big)^p$. Then the $p$th power of the $p$-Wasserstein distance between the measures $\mu$ and $\nu$ is the optimal value of the following transportation problem:

$$\min_{\pi \in R_+^{N \times M}} \sum_{i=1}^N \sum_{k=1}^M d_{ik} \pi_{ik} \quad \text{s.t.} \quad \pi^\top \mathbf{1}_N = w_x, \quad \pi \mathbf{1}_M = w_z. \tag{6}$$

The calculation of the distance is easy when the linear programming problem (6) can be solved. For large instances, specialized algorithms such as (Cuturi, 2013; Genevay et al., 2016; Altschuler et al., 2017; Lin et al., 2019; Dvurechensky et al., 2018; Dvurechenskii et al., 2018; Kroshnin et al., 2019; Cuturi & Peyré, 2018; 2016) have been proposed. Our problem, in this special case, is more complex: *find $\nu$ supported on a set of the cardinality $M$ such that $W_p(\mu,\nu)$ is the smallest possible.* We elaborate on it in the next section.

### 2.2 KERNEL DISTANCE

To define a metric between kernels, we restrict the class of kernels under consideration to the set $\mathscr{Q}_p(\mathscr{X})$ of kernels $Q : \mathscr{X} \to \mathscr{P}_p(\mathscr{X})$ such that for each a constant $C$ exists, with which

$$\int_{\mathscr{X}} d(y,y_0)^p \, Q(\mathrm{d}y|x) \leq C\big(1 + d(x,x_0)^p\big), \quad \forall x \in \mathscr{X}.$$

The choice of the points $x_0 \in \mathscr{X}$ and $y_0 \in \mathscr{Y}$ is irrelevant, because $C$ may be adjusted.

**Definition 2.1** *The transportation distance of order $p$ between two kernels $Q$ and $\widetilde{Q}$ in $\mathscr{Q}_p(\mathscr{X})$ with a fixed marginal $\lambda \in \mathscr{P}_p(\mathscr{X})$ is defined as*

$$\mathscr{W}_p^\lambda(Q,\widetilde{Q}) = \left( \int_{\mathscr{X}} \big[ W_p(Q(\cdot|x),\widetilde{Q}(\cdot|x)) \big]^p \, \lambda(\mathrm{d}x) \right)^{1/p}.$$

For a fixed marginal $\lambda \in \mathscr{P}_p(\mathscr{X})$, we identify the kernels $Q$ and $\widetilde{Q}$ if $W_p(Q(\cdot|x),\widetilde{Q}(\cdot|x)) = 0$ for $\lambda$-almost all $x \in \mathscr{X}$. In this way, we define the space $\mathscr{Q}_p^\lambda(\mathscr{X},\mathscr{Y})$ of equivalence classes of $\mathscr{Q}_p(\mathscr{X},\mathscr{Y})$.

**Theorem 2.1** *For any $p \in [1,\infty)$ and any $\lambda \in \mathscr{P}_p(\mathscr{X})$, the function $\mathscr{W}_p^\lambda(\cdot,\cdot)$, defines a metric on the space $\mathscr{Q}_p^\lambda(\mathscr{X},\mathscr{Y})$.*

The proof is provided in Appendix A.1.1.

The kernel distance can be used to approximate the system (1) by a system with finitely supported kernels. Suppose at stage $t$ we already have for all $\tau = 0, \ldots, t-1$ approximate kernels $\widetilde{Q}_\tau : \mathscr{X} \to \mathscr{P}(\mathscr{X})$. These kernels define the approximate marginal distribution

$$\widetilde{\lambda}_t = \delta_{x_0} \circ \widetilde{Q}_0 \circ \widetilde{Q}_1 \circ \cdots \circ \widetilde{Q}_{t-1} = \widetilde{\lambda}_{t-1} \circ \widetilde{Q}_{t-1}.$$

We also have the finite subsets $\mathscr{X}_\tau = \mathrm{supp}(\widetilde{\lambda}_\tau)$, $\tau = 0, 1, \ldots, t$. For $t = 0$, $\widetilde{\lambda}_0 = \delta_{x_0}$, and $\mathscr{X}_0 = \{x_0\}$.

At the stage $t$, we construct a kernel $\widetilde{Q}_t : \mathscr{X}_t \to \mathscr{P}_p(\mathscr{X})$ such that

$$\mathscr{W}_p^{\widetilde{\lambda}_t}(Q_t, \widetilde{Q}_t) \le \Delta_t. \tag{6}$$

If $t < T - 1$, we increase $t$ by one, and continue; otherwise, we stop. Observe that the approximate marginal distribution $\widetilde{\lambda}_t$ is well-defined at each step of this abstract scheme.

We then solve the approximate version of the risk evaluation algorithm (2), with the true kernels $Q_t$ replaced by the approximate kernels $\widetilde{Q}_t$, $t = 0, \ldots, T-1$:

$$\widetilde{v}_t(x) = c_t(x) + \sigma_t\big(x, \widetilde{Q}_t(x), \widetilde{v}_{t+1}(\cdot)\big), \quad x \in \mathscr{X}_t, \quad t = 0, 1, \ldots, T-1; \tag{7}$$

we assume that $\widetilde{v}_T(\cdot) \equiv v_T(\cdot) \equiv c_T(\cdot)$.

To estimate the error of this evaluation in terms of the kernel errors $\Delta_t$, we make the following general assumptions.

(A) For every $t = 0, 1, \ldots, T-1$ and for every $x \in \mathscr{X}_t$, the operator $\sigma_t(x, \cdot, v_{t+1})$ is Lipschitz continuous with respect to the metric $W_p(\cdot, \cdot)$ with the constant $L_t$:

$$\big|\sigma_t\big(x, \mu, v_{t+1}(\cdot)\big) - \sigma_t\big(x, \nu, v_{t+1}(\cdot)\big)\big| \le L_t W_p(\mu, \nu), \quad \forall \mu, \nu \in \mathscr{P}_p(\mathscr{X});$$

(B) For every $x \in \mathscr{X}_t$ and for every $t = 0, 1, \ldots, T-1$, the operator $\sigma_t(x, \widetilde{Q}_t(x), \cdot)$ is Lipschitz continuous with respect to the norm in the space $\mathscr{L}_p(\mathscr{X}, \mathscr{B}(\mathscr{X}), \widetilde{Q}_t(x))$ with the constant $K_t$:

$$\big|\sigma_t\big(x, \widetilde{Q}_t(x), v(\cdot)\big) - \sigma_t\big(x, \widetilde{Q}_t(x), w(\cdot)\big)\big| \le K_t \|v - w\|_p, \quad \forall v, w \in \mathscr{L}_p(\mathscr{X}, \mathscr{B}(\mathscr{X}), \widetilde{Q}_t(x)).$$

**Theorem 2.2** *If the assumptions* (A) *and* (B) *are satisfied, then for all* $t = 0, \ldots, T-1$ *we have*

$$\left(\int_{\mathscr{X}} |\widetilde{v}_t(x) - v_t(x)|^p \, \widetilde{\lambda}_t(dx)\right)^{1/p} \le \sum_{\tau=t}^{T-1} L_\tau \left(\prod_{j=t}^{\tau-1} K_j\right) \Delta_\tau. \tag{8}$$

The proof is provided in Appendix A.1.2.

In order to accomplish (6), at stage $t$, we construct a finite set $\mathscr{X}_{t+1} \subset \mathscr{X}$ of cardinality $M_{t+1}$ and a kernel $\widetilde{Q}_t : \mathscr{X}_t \to \mathscr{P}(\mathscr{X}_{t+1})$ by solving the following problem:

$$\begin{aligned} \min_{\mathscr{X}_{t+1}, \widetilde{Q}_t} \quad & \mathscr{W}_p^{\widetilde{\lambda}_t}(Q_t, \widetilde{Q}_t) \\ \text{s.t. } & \mathrm{supp}(\widetilde{\lambda}_t \circ \widetilde{Q}_t) = \mathscr{X}_{t+1}, \\ & |\mathscr{X}_{t+1}| \le M_{t+1}. \end{aligned} \tag{9}$$

The cardinality $M_{t+1}$ has to be chosen experimentally, to achieve the desired accuracy in (6). After (approximately) solving this problem, we increase $t$ by one and continue.

Let us focus on effective ways for constructing an approximate solution to problem (9). We represent the (unknown) support of $\widetilde{\lambda}_t \circ \widetilde{Q}_t$ by $\mathscr{X}_{t+1} = \{z_{t+1}^\ell\}_{\ell=1, \ldots, M_{t+1}}$ and the (unknown) transition probabilities by $\widetilde{Q}_t(z_{t+1}^\ell | z_t^s)$, $s = 1, \ldots, M_n$, $\ell = 1, \ldots, M_{n+1}$. With the use of the kernel distance, problem (9) can be equivalently rewritten as:

$$\begin{aligned} \min_{\mathscr{X}_{t+1}, \widetilde{Q}_t} \quad & \sum_{s=1}^{M_n} \widetilde{\lambda}_t^s W_p\big(Q_t(\cdot | z_t^s), \widetilde{Q}_t(\cdot | z_t^s)\big)^p \\ \text{s.t. } & \mathrm{supp}\big(\widetilde{Q}_t(\cdot | z_t^s)\big) \subset \mathscr{X}_{t+1}, \quad s = 1, \ldots, M_n, \\ & |\mathscr{X}_{t+1}| \le M_{t+1}. \end{aligned} \tag{10}$$

In our approach, we represent each distribution $Q_t(\cdot|z_t^s)$ by a finite number of particles $\{x_{t+1}^{s,i}\}_{i \in \mathscr{I}_{t+1}^s}$ drawn independently from $Q_t(\cdot|z_t^s)$. The expected error of this approximation is well-investigated by Dereich et al. (2013) and Fournier & Guillin (2015) in terms of the sample size $|\mathscr{I}_{t+1}^s|$, the state space dimension, and the distribution's moments. Assuming the error of this large-size discrete approximation as fixed, we aim to construct a smaller support with as little error as possible to the particle distribution. For this purpose, we introduce the sets $\mathscr{Z}_{t+1} = \{\zeta_{t+1}^k\}_{k=1,\ldots,K_{t+1}}$. Each consists of pre-selected potential locations for the next-stage representative states $z_{t+1}^j$, where $j = 1,\ldots,M_{t+1}$. It may be the union of the sets of particles, $\{x_{t+1}^{s,i}, i \in \mathscr{I}_{t+1}^s, s = 1,\ldots,M_t\}$; often, computational expediency requires that $K_{t+1} < \sum_{s=1}^{M_t} |\mathscr{I}_{t+1}^s|$, we still have $M_{t+1} \ll K_{t+1}$, which makes the task of finding the best representative points challenging.

If the next-stage representative points $\{z_{t+1}^j\}_{j=1,\ldots,M_{t+1}}$ were known, the problem would have a straightforward solution. For each particle $x_{t+1}^{s,i}$ we would choose the closest representative point, $j^*(i) = \arg\min_{j=1,\ldots,M_{t+1}} d(x_{t+1}^{s,i}, z_{t+1}^j)$, and set the transportation probabilities $\pi_t^{s,i,j^*(k)} = \frac{1}{|\mathscr{I}_{t+1}^s|}$; for other $j$, we set them to 0. The implied approximate kernel is $\widetilde{Q}_t(z_{t+1}^j|z_t^s) = \sum_{i \in \mathscr{I}_{t+1}^s} \pi_t^{s,i,j}$, $s = 1,\ldots,M_t$, $j = 1,\ldots,M_{t+1}$; it is the proportion of the particles from $\mathscr{I}_{t+1}^s$ assigned to $z_{t+1}^j$.

To find the best representative points, we introduce the binary variables

$$\gamma_k = \begin{cases} 1 & \text{if the point } \zeta_{t+1}^k \text{ has been selected to } \mathscr{X}_{t+1}, \\ 0 & \text{otherwise}, \end{cases} \quad k = 1,\ldots,K_{t+1},$$

and we re-scale the transportation plans:

$$\beta_{sik} = |\mathscr{I}_{t+1}^s|\pi_t^{s,i,k}, \quad s = 1,\ldots,M_t, \ i \in \mathscr{I}_{t+1}^s, \ k = 1,\ldots,K_{t+1}.$$

We obtain from (10) the following linear mixed-integer optimization problem (we omit the ranges of the sums when they are evident):

$$\min_{\gamma,\beta} \sum_s w_s \sum_i \sum_k d_{sik}\beta_{sik} \tag{11a}$$

$$\text{s.t. } \beta_{sik} \in [0,1], \ \gamma_k \in \{0,1\}, \quad s = 1,\ldots,M_t, \quad i \in \mathscr{I}_{t+1}^s, \quad k = 1,\ldots,K_{t+1}, \tag{11b}$$

$$\beta_{sik} \leq \gamma_k, \quad s = 1,\ldots,M_t, \quad i \in \mathscr{I}_{t+1}^s, \quad k = 1,\ldots,K_{t+1}, \tag{11c}$$

$$\sum_k \beta_{sik} = 1, \quad s = 1,\ldots,M_t, \quad i \in \mathscr{I}_{t+1}^s, \tag{11d}$$

$$\sum_k \gamma_k \leq M_{t+1}, \tag{11e}$$

with $w_s = \frac{\widetilde{\lambda}_s^s}{|\mathscr{I}_{t+1}^s|}$ and $d_{sik} = d(x_{t+1}^{s,i}, \zeta_{t+1}^k)^p$. The implied approximate kernel is:

$$\widetilde{Q}_t(z_{t+1}^k|z_t^s) = \frac{1}{|\mathscr{I}_{t+1}^s|} \sum_i \beta_{sik}, \quad s = 1,\ldots,M_t, \quad k = 1,\ldots,M_{t+1}. \tag{12}$$

Finally, $\widetilde{\lambda}_{t+1} = \widetilde{\lambda}_t \circ \widetilde{Q}_t$, and the iteration continues until $t = T - 1$.

Since problem (11) involves binary variables, it is reasonable to employ an integer programming solver, such as Gurobi, CPLEX, or SCIP. However, integer or even linear programming can become computationally intractable for large-scale problems with many variables and constraints. Therefore, in section 3, we propose a subgradient-based method to solve problem (11).

The particle selection problem using the Wasserstein distance is a simplified form of problem (11). In the case of $M_t = 1$, we obtain the problem of finding the best $v$ in (6). Notably, the facility location and clustering problems share similarities with our particle selection method as well.

## 3 DUAL SUBGRADIENT METHOD

In this section, we propose a subgradient algorithm to address the computational intractability of large-scale instances of problem (11). While the subgradient method does not ensure convergence

to the strictly optimal solution, it is faster than the mixed-integer linear programming approach and it scales better. We use the fact that our primary objective is to determine the $\gamma$'s, which the subgradient method can effectively accomplish. We present the dual problem in Section 3.1, and the exact algorithm used for selecting particles in Section 3.2.

## 3.1 THE DUAL PROBLEM

Assigning Lagrange multipliers $\theta_{si}$ and $\theta_0 \geq 0$ to the constrains (11d) and (11e), respectively, we obtain the Lagrangian function of problem (11):

$$L(\gamma, \beta; \theta) = \sum_s \sum_i \sum_k w_s d_{sik} \beta_{sik} + \sum_s \sum_i \theta_{si} \left(1 - \sum_k \beta_{sik}\right) + \theta_0 \left(\sum_k \gamma_k - M_{t+1}\right).$$

The dual variable $\theta_0$ has the interpretation of the marginal contribution of an additional point to reducing the kernel distance. The variables $\theta_{si}$ serve as thresholds in the assignment of the particles $x_{t+1}^{s,i}$ to the candidate points. They are needed for the algorithm but are not used in the final assignment, which can be done easily once the $\gamma$'s are known. The corresponding dual function is

$$
\begin{aligned}
L_D(\theta) &= \min_{\gamma, \beta \in \Gamma} L(\gamma, \beta; \theta) \\
&= \sum_{k=1}^{K_{t+1}} \left\{ \min_{\gamma_k, \beta_{\cdot\cdot k} \in \Gamma_k} \sum_{s=1}^{M_t} \sum_{i \in \mathscr{I}_{t+1}^s} (w_s d_{sik} - \theta_{si}) \beta_{sik} + \theta_0 \gamma_k \right\} + \sum_{s=1}^{M_t} \sum_{i \in \mathscr{I}_{t+1}^s} \theta_{si} - M_{t+1} \theta_0,
\end{aligned}
\tag{13}
$$

where $\Gamma$ is the feasible set of the primal variables given by the conditions (11b)–(11c), and $\Gamma_k$ is its projection on the subspace associated with the $k$th candidate point $\zeta_{t+1}^k$. The minimization in (13) decomposes into $K_{t+1}$ subproblems, each having a closed-form solution. We can perform these calculations in parallel, which provides a significant computational advantage and reduces the optimization time. We see that $\beta_{sik} = 1$, if $\gamma_k = 1$ and $\theta_{si} > w_s d_{sik}$; it may be arbitrary in $[0,1]$, if $\gamma_k = 1$ and exact equality is satisfied; and is 0, otherwise. Therefore, for all $k = 1, \ldots, K_{t+1}$,

$$\gamma_k = 1, \quad \text{if} \quad \theta_0 < \sum_{s=1}^{M_t} \sum_{i \in \mathscr{I}_{t+1}^s} \max(0, \theta_{si} - w_s d_{sik});$$

$\gamma_k \in \{0,1\}$, if exact equality holds; and $\gamma_k = 0$, otherwise. We denote by $\hat{\Gamma}(\theta)$ the set of solutions of problem (13). It is worth stressing that in the algorithm below, we need only *one* solution for each $\theta$.

The dual problem has the form

$$\max_\theta L_D(\theta), \quad \text{s.t.} \quad \theta_0 \geq 0. \tag{14}$$

The optimal value of (14) may be strictly below the optimal value of (11); it is equal to the optimal value of the linear programming relaxation, where the conditions $\gamma_k \in \{0,1\}$ are replaced by $\gamma_k \in [0,1]$. However, if we replace $M_{t+1}$ by the number of $\gamma_k$'s equal to 1, the gap is zero. If we keep $M_{t+1}$ unchanged, we can construct a feasible solution by setting to 0 the $\gamma_k$'s for which the change in the expression in the braces in (13) is the smallest. This allows for estimation of the gap.

The subdifferential of the dual function has the form

$$\partial L_D(\theta) = \text{conv} \left\{ \begin{bmatrix} \left\{1 - \sum_{k=1}^K \hat{\beta}_{sik}\right\}_{s=1,\ldots,M_t,\ i \in \mathscr{I}_{t+1}^s} \\ \sum_{k=1}^K \hat{\gamma}_k - M_{t+1} \end{bmatrix} : (\hat{\gamma}, \hat{\beta}) \in \hat{\Gamma}(\theta) \right\}. \tag{15}$$

At the optimal solution $\hat{\theta}$ we have $0 \in \partial L_D(\hat{\theta})$, because $\hat{\theta}_0 > 0$ (the constraint (11e) must be active).

## 3.2 THE ALGORITHM

In Algorithm 1, we use $j$ to denote the iteration number, starting from 0. The variable $\theta$ represents the initial values of the dual variables, while $M$ represents the number of desired grid points. The parameter $\varepsilon$ specifies the tolerance level. The value $\alpha^{(0)}$ denotes the initial learning rate. The variables $\varkappa_1$ and $\varkappa_2$ are exponential decay factors between 0 and 1, which determine the relative contribution of the current gradient and earlier gradients to the direction. It is important to note that the total number of $\gamma$'s selected by the subgradient method may not necessarily equal $M$, when the

stopping criteria are met. However, for the particle selection method, the constraint $\sum_{k=1}^{K} \gamma_k \leq M_{t+1}$ is not strictly enforced (it is a modeling issue). We end the iteration when $\sum_{k=1}^{K} \gamma_k$ is close to $M_{t+1}$.

For the (approximate) primal recovery, we choose $J$ last values $\theta^{(j)}$ at which $L_D(\theta^{(j)})$ is near optimal, and consider the convex hull of the observed subgradients of the dual function at these points as an approximation of the subdifferential (15). The minimum norm element in this convex hull corresponds to a convex combination of the corresponding dual points: $(\bar{\gamma}, \bar{\beta}) = \sum_{j \in J} \omega_j (\gamma^{(j)}, \beta^{(j)})$, with $\sum_{j \in J} \omega_j = 1$, $\omega_j \geq 0$.

By the duality theory in convex optimization, if the subgradients were collected at the optimal point, $(\bar{\gamma}, \bar{\beta})$ would be the solution of the convex relaxation of (11). So, if the norm of the convex combination of the subgradients is small, then $\sum_{k=1}^{K} \bar{\gamma}_k \approx M_{t+1}$, and we may regard $\bar{\gamma}$ as an approximate solution. We interpret it as the best "mixed strategy" and select each point $k$ with probability $\bar{\gamma}_k$. In our experiments, we simply use $\omega_j = \left( \sum_{i \in J} \alpha^{(i)} \right)^{-1} \alpha^{(j)} \approx 1/|J|$. This approach is well supported theoretically by (Larsson et al., 1999). The $\mathscr{O}(1/\sqrt{j+1})$ rate of convergence of the subgradient method is well understood since (Zinkevich, 2003) (see also the review by Garrigos & Gower (2023)).

---

**Algorithm 1:** Dual subgradient method with momentum

**Input** : $\theta^{(0)}, M, \varepsilon, \alpha^{(0)}, \varkappa_1, \varkappa_2$ and $j = 0$.
**Output :** $\theta, \gamma$, and $\beta$.

1 **while** $\sum_{k=1}^{K} \gamma_k < (1-a) * M$ **or** $\sum_{k=1}^{K} \gamma_k > (1+a) * M$ **or** $\|L_D(\theta^{(j)}) - L_D(\theta^{(j-1)})\| > \varepsilon$ **do**
2     **for** $k = 1, \dots, K$ **do**
3        **if** $\sum_{s=1}^{N} \sum_{i \in \mathscr{I}^s} \max(0, \theta_{si} - w_s d_{sik}) > \theta_0$ **then**
4           $\gamma_k \leftarrow 1$;
5           $\beta_{sik} \leftarrow \mathbf{1}_{\{w_s d_{sik} < \theta_{si}^{(j)}\}}, \quad s = 1, \dots, N, i \in \mathscr{I}^s$;
6        **else**
7           $\gamma_k \leftarrow 0$;
8           $\beta_{sik} \leftarrow 0, \quad s = 1, \dots, N, i \in \mathscr{I}^s$;
9        **end**
10     **end**
11     $\alpha^{(j+1)} \leftarrow \frac{\alpha^{(0)}}{\sqrt{j+1}}$;
12     $m_0^{(j+1)} \leftarrow (1 - \varkappa_1)(\sum_{k=1}^{K} \gamma_k - M) + \varkappa_1 m_0^{(j)}$;
13     $\theta_0^{(j+1)} \leftarrow \theta_0^{(j)} + \alpha^{(j+1)} m_0^{(j+1)}$;
14     $m_{si}^{(j+1)} \leftarrow (1 - \varkappa_2)(1 - \sum_{k=1}^{K} \beta_{sik}) + \varkappa_2 m_{si}^{(j)}$;
15     $\theta_{si}^{(j+1)} \leftarrow \theta_{si}^{(j)} + \alpha^{(j+1)} m_{si}^{(j+1)} \quad s = 1, \dots, N, i \in \mathscr{I}^s$;
16     $j \leftarrow j + 1$
17 **end**

---

In the stochastic version of the method, the loop over $k$ in lines 2–10 is executed in a randomly selected batch $\mathscr{B}^{(j)} \subset \{1, \dots, K\}$ of size $B \ll K$. Then, in line 12, the subgradient component $g_0^{(j)} = \sum_{k=1}^{K} \gamma_k - M$ is replaced by its stochastic estimate $\tilde{g}_0^{(j)} = (K/B) \sum_{k \in \mathscr{B}^{(j)}} \gamma_k - M$. In line 14, the subgradient components $g_{si}^{(j)} = 1 - \sum_{k=1}^{K} \beta_{sik}$ are replaced by their estimates $\tilde{g}_{si}^{(j)} = 1 - (K/B) \sum_{k \in \mathscr{B}^{(j)}} \beta_{sik}$. If the batches are independently drawn at each iteration, the algorithm is a version of the stochastic subgradient method with momentum (see (Yan et al., 2018; Liu et al., 2020) and the references therein).

## 4 NUMERICAL ILLUSTRATION - THE OPTIMAL STOPPING PROBLEM

Consider an $n$-dimensional stochastic process $\{S_t^{(i)}\}$, $i = 1, \dots, n$, following (under a probability measure $\mathbb{Q}$) a geometric Brownian motion:

$$\frac{\mathrm{d}S_t^{(i)}}{S_t^{(i)}} = r\,\mathrm{d}t + \sigma^{(i)}\,\mathrm{d}W_t^{\mathbb{Q}}, \quad i = 1, \dots, n, \quad t \in [0, T]. \tag{16}$$

Here, $\{W_t^{\mathbb{Q}}\}$ is an $n$-dimensional Brownian motion under probability measure $\mathbb{Q}$, $r$ is a constant coefficient, and $\sigma^{(i)}$ is the $n$ dimensional (row) vector coefficients of $S^{(i)}$.

We examine an optimal stopping risk function associated with this stochastic process. If we stop the process at time t, the reward is $\Phi(S_t)$, where $\Phi : \mathbb{R}^n \to [0, +\infty)$ is a known function. The problem is to design a stopping strategy that maximizes the expected reward. The optimal value of this stopping problem is:

$$V_t(x) = \sup_{\substack{\tau-\text{stopping time} \\ t \le \tau \le T}} E^{\mathbb{Q}}\left[e^{-r(\tau-t)}\Phi(S_\tau) \big| S_t = x\right], \quad x \in \mathbb{R}^n. \tag{17}$$

To develop a numerical scheme for approximating this value, we first partition the time interval $[0, T]$ into short intervals of length $\Delta t = T/N$, defining the set $\Gamma_N = \{t_i = i\Delta t : i = 0, 1, \dots, N\}$. With the exercise times restricted to $\Gamma_N$, we approximate the value function by

$$V_t^{(N)}(x) = \sup_{\substack{\tau-\text{stopping time} \\ \tau \in \Gamma_N}} E^{\mathbb{Q}}\left[e^{-r(\tau-t)}\Phi(S_\tau) \big| S_t = x\right], \quad t \in \Gamma_N, \quad x \in \mathbb{R}^n. \tag{18}$$

We view $V_t^{(N)}(x)$ as an approximation to the actual risk measure (17) when $N$ is sufficiently large. It satisfies the following dynamic programming equations:

$$V_{t_N}^{(N)}(x) = \Phi(x), \quad x \in \mathbb{R}^n,$$

$$V_{t_i}^{(N)}(x) = \max\left\{\Phi(x), E^{\mathbb{Q}}\left[e^{-r\Delta t}V_{t_{i+1}}^{(N)}(S_{t_{i+1}}) \big| S_{t_i} = x\right]\right\}, \quad i = 0, 1, \dots, N-1,$$

which are a special case of the backward system (2).

We evaluated the performance of two methods for simulating the stochastic process' movements and estimating the values of the risk measure. The first method is the grid point selection method which relies on the kernel distance. At each time step $t_i$, we choose the representative point(s) $z_i^j, j = 1, \dots, M_i$ to represent the state space. The pre-selected potential locations of the representative particles are simulated from the true distribution as well. Since we have set the total number of time intervals to $N = 30$, the grid point selection algorithm needs to be executed 30 times. Due to the large number of variables, with a size of 31727 selected grid points at $N = 30$ alone, the MIP solver is extremely inefficient and takes several days to converge. In contrast, Algorithm 1 takes only a few hours to select grid points, making it the only viable option.

We compared this method with the classical binomial tree approach in which the Brownian motion is approximated by an $n$-dimensional random walk on a binomial tree. It may be complex and computationally expensive, particularly for large numbers of time steps, since the total number of nodes of the tree grows exponentially with the dimension $n$ and time. Our method based on the kernel distance reduces the number of nodes or grid points, especially at later stages, while still providing accurate results. Additionally, it can accommodate a wide range of stochastic processes, whereas the binomial tree method is limited to log-normal distributions.

Both methods were employed to evaluate two reward functions with $n = 3$, such that the assumptions for Theorem (2.2) are satisfied. The first reward function is denoted by $\Phi_p(S_t) = \max\left(K - \sum_{i=1}^n w_i S_t^{(i)}, 0\right)$, while the second reward function is represented by $\Phi_{mp}(S_t) = \max\left(K - \max_{i=1,\dots,n}(S_t^{(i)}), 0\right)$. Here, $w_i$ represents the percentage of variable $i$, and $K$ is a constant coefficient. The parameter values used were $S_0 = [5, 10, 8]$, $r = 0.03$, $K = 8$, $w = (\frac{1}{3}, \frac{1}{3}, \frac{1}{3})$, and $T = 1$. The $\sigma$ were:

$$\sigma = \begin{bmatrix} 0.5 & -0.2 & -0.1 \\ -0.2 & 1 & 0.3 \\ -0.1 & 0.3 & 0.8 \end{bmatrix}.$$

In Table 1, the approximated values using the grid point selection method and the binomial tree method are compared. Additionally, Figures 1a and 1b present the convergence of the reward functions as the total number of time discretization steps increases.

Table 1: The reward functions for different time discretization steps.

| $N$ | $\Phi_p$ - grid | $\Phi_p$ - binomial | $\Phi_{mp}$ - grid | $\Phi_{mp}$ - binomial |
|---|---|---|---|---|
| 1 | 1.974 | 1.921 | 0.786 | 0.530 |
| 2 | 1.982 | 2.008 | 0.949 | 1.066 |
| 3 | 1.994 | 1.998 | 1.024 | 1.016 |
| 5 | 2.000 | 1.974 | 1.087 | 1.053 |
| 6 | 2.003 | 1.980 | 1.111 | 1.077 |
| 10 | 1.994 | 2.001 | 1.145 | 1.163 |
| 15 | 1.992 | 2.000 | 1.172 | 1.178 |
| 30 | 2.004 | 2.002 | 1.217 | 1.222 |

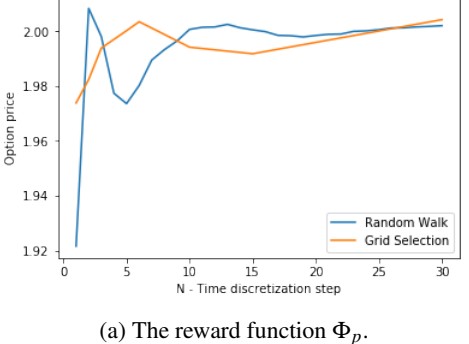

(a) The reward function $\Phi_p$.

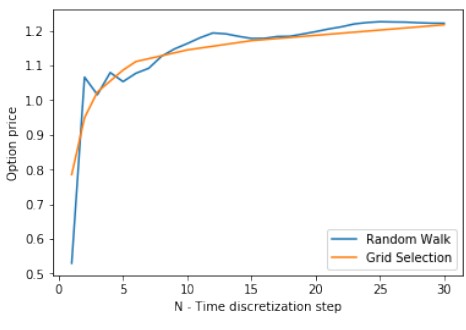

(b) The reward function $\Phi_{mp}$.

Figure 1: The approximate value of the reward functions vs. the number of time discretization steps

## 5    CONCLUSION

We introduced a kernel distance metric based on the Wasserstein distance between probability distributions and considered a new problem of approximating a large-scale Markov system with a simpler system and a finite state space. For this problem, we proposed a novel particle selection method that iteratively approximates the forward system stage-by-stage by utilizing our kernel distance. The heart of the method is a decomposable and parallelizable subgradient algorithm for particle selection, designed to circumvent the complexities of dealing with constraints and matrix computations.

To empirically validate our approach, we conducted extensive experiments and applied our methodology to the optimal stopping problem. We benchmarked our results against the binomial tree method, recognized as the state-of-the-art technique for approximating geometric Brownian motion. Furthermore, in Appendix A.2, we provide a straightforward example involving a 2-dimensional and 1-time-stage Gaussian distribution. We selected this simple case to aid in visualizing outcomes, enabling effective method comparisons and highlighting the limitations of Mixed Integer Programming (MIP) solvers in more complicated scenarios.

Additionally, it's worth noting that the kernel distance and the particle selection method hold significant potential for various applications. One such application pertains to look-ahead risk assessment in reinforcement learning, specifically in the context of Markov risk measures. Evaluating Markov risk measures in dynamic systems can be achieved through the equation (2), offering superior performance over one-step look-ahead methods. Our approach streamlines risk or reward evaluation across a range of scenarios by substituting the approximate kernel in place of the original equation (2). We intend to explore the full spectrum of potential applications for our work in future research endeavors.

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

# A  APPENDIX

## A.1  PROOFS OF THEOREMS

### A.1.1  PROOF OF THEOREM 2.1

It is obvious that $\mathscr{W}_p^\lambda(Q,\widetilde{Q}) \geq 0$ for any $Q,\widetilde{Q} \in \mathscr{Q}_p^\lambda(\mathscr{X},\mathscr{Y})$ and $\mathscr{W}_p^\lambda(Q,\widetilde{Q}) = 0$ if and only if $Q = \widetilde{Q}$ $\lambda$-a.s.. We next verify the triangle inequality. For all $Q,Q',\widetilde{Q} \in \mathscr{Q}_p^\lambda(\mathscr{X},\mathscr{Y})$, by the triangle inequality for $W_p(\cdot,\cdot)$ and then by the Minkowski inequality, we obtain

$$\mathscr{W}_p^\lambda(Q,\widetilde{Q}) \leq \left( \int_{\mathscr{X}} \left[ W_p(Q(\cdot|x),Q'(\cdot|x)) + W_p(Q'(\cdot|x),\widetilde{Q}(\cdot|x)) \right]^p \lambda(\mathrm{d}x) \right)^{1/p}$$

$$\leq \left( \int_{\mathscr{X}} \left[ W_p(Q(\cdot|x),Q'(\cdot|x)) \right]^p \lambda(\mathrm{d}x) \right)^{1/p} + \left( \int_{\mathscr{X}} \left[ W_p(Q'(\cdot|x),\widetilde{Q}(\cdot|x)) \right]^p \lambda(\mathrm{d}x) \right)^{1/p}$$

$$= \mathscr{W}_p^\lambda(Q,Q') + \mathscr{W}_p^\lambda(Q',\widetilde{Q}).$$

Furthermore, setting $Q'(\cdot|x) = \delta_{\{y_0\}}(\cdot)$, we get

$$
\begin{aligned}
\left[ \mathscr{W}_p^\lambda(Q,\delta_{\{y_0\}}) \right]^p &= \int_{\mathscr{X}} \left[ W_p(Q(\cdot|x),\delta_{\{y_0\}}) \right]^p \lambda(\mathrm{d}x) \\
&= \int_{\mathscr{X}} \int_{\mathscr{Y}} d(y,y_0)^p \, Q(\mathrm{d}y|x) \, \lambda(\mathrm{d}x) \leq C(Q) \int_{\mathscr{X}} \left( 1 + d(x,x_0)^p \right) \lambda(\mathrm{d}x) < \infty,
\end{aligned}
\tag{19}
$$

which proves the finiteness of $\mathscr{W}_p^\lambda(Q,\widetilde{Q})$, if $\lambda \in \mathscr{P}_p(\mathscr{X})$.

### A.1.2  PROOF OF THEOREM 2.2

First, we prove by induction backward in time that for all $t = 0,1,\ldots,T-1$ and all $x \in \mathscr{X}_t$ we have

$$\left| \widetilde{v}_t(x) - v_t(x) \right| \leq \sum_{\tau=t}^{T-1} L_\tau \left( \prod_{j=t}^{\tau-1} K_j \right) \mathscr{W}_p^{\delta_x \circ \widetilde{Q}_t \circ \cdots \circ \widetilde{Q}_{\tau-1}}(\widetilde{Q}_\tau, Q_\tau). \tag{20}$$

At the time $t = T-1$, assumption (A) yields the inequality

$$
\begin{aligned}
\left| \widetilde{v}_{T-1}(x) - v_{T-1}(x) \right| &\leq \left| \sigma_{T-1}\left( x, \widetilde{Q}_{T-1}(x), v_T(\cdot) \right) - \sigma_{T-1}\left( x, Q_{T-1}(x), v_T(\cdot) \right) \right| \\
&\leq L_{T-1} W_p(\widetilde{Q}_{T-1}(x), Q_{T-1}(x)) = L_{T-1} \mathscr{W}_p^{\delta_x}(\widetilde{Q}_{T-1}, Q_{T-1}),
\end{aligned}
$$

which is the same as (20) for $T-1$. Supposing (20) is true for $t$, we verify it for $t-1$. Using assumptions (A) and (B) we obtain:

$$
\begin{aligned}
&\left| \widetilde{v}_{t-1}(x) - v_{t-1}(x) \right| \\
&\leq \left| \sigma_{t-1}\left( x, \widetilde{Q}_{t-1}(x), v_t(\cdot) \right) - \sigma_{t-1}\left( x, Q_{t-1}(x), v_t(\cdot) \right) \right| \\
&\quad + \left| \sigma_{t-1}\left( x, \widetilde{Q}_{t-1}(x), \widetilde{v}_t(\cdot) \right) - \sigma_{t-1}\left( x, \widetilde{Q}_{t-1}(x), v_t(\cdot) \right) \right| \\
&\leq L_{t-1} W_p(\widetilde{Q}_{t-1}(x), Q_{t-1}(x)) + K_{t-1} \left( \int_{\mathscr{X}} \left| \widetilde{v}_t(y) - v_t(y) \right|^p \widetilde{Q}_{t-1}(\mathrm{d}y|x) \right)^{1/p}.
\end{aligned}
$$

The substitution of (20) and the application of the Minkowski inequality yield

$$
\begin{aligned}
\left| \widetilde{v}_{t-1}(x) - v_{t-1}(x) \right| &\leq L_{t-1} \mathscr{W}_p^{\delta_x}(\widetilde{Q}_{t-1}, Q_{t-1}) \\
&\quad + K_{t-1} \sum_{\tau=t}^{T-1} L_\tau \left( \prod_{j=t}^{\tau-1} K_j \right) \left( \int_{\mathscr{X}} \left[ \mathscr{W}_p^{\delta_y \circ \widetilde{Q}_t \circ \cdots \circ \widetilde{Q}_{\tau-1}}(\widetilde{Q}_\tau, Q_\tau) \right]^p \widetilde{Q}_{t-1}(\mathrm{d}y|x) \right)^{1/p}.
\end{aligned}
$$

Observing that

$$\int_{\mathscr{X}} \left[ \mathscr{W}_p^{\delta_y \circ \widetilde{Q}_t \circ \cdots \circ \widetilde{Q}_{\tau-1}}(\widetilde{Q}_\tau, Q_\tau) \right]^p \widetilde{Q}_{t-1}(\mathrm{d}y|x) = \left[ \mathscr{W}_p^{\delta_x \circ \widetilde{Q}_{t-1} \circ \widetilde{Q}_t \circ \cdots \circ \widetilde{Q}_{\tau-1}}(\widetilde{Q}_\tau, Q_\tau) \right]^p, \tag{21}$$

we can write the preceding displayed inequality as

$$\left|\widetilde{v}_{t-1}(x) - v_{t-1}(x)\right| \leq L_{t-1}\mathscr{W}_p^{\delta_x}(\widetilde{Q}_{t-1}, Q_{t-1}) + K_{t-1}\sum_{\tau=t}^{T-1} L_\tau\left(\prod_{j=t}^{\tau-1} K_j\right)\mathscr{W}_p^{\delta_x \circ \widetilde{Q}_{t-1} \circ \widetilde{Q}_t \circ \cdots \circ \widetilde{Q}_{\tau-1}}(\widetilde{Q}_\tau, Q_\tau),$$

which is the same as (20) for $t-1$. By induction, (20) is true for all $t$.

The formula (8) follows now by integrating the right-hand side of (20) and using the identity

$$\int_{\mathscr{X}} \left[\mathscr{W}_p^{\delta_x \circ \widetilde{Q}_t \circ \cdots \circ \widetilde{Q}_{\tau-1}}(\widetilde{Q}_\tau, Q_\tau)\right]^p \widetilde{\lambda}_t(\mathrm{d}x) = \left[\mathscr{W}_p^{\widetilde{\lambda}_\tau}(\widetilde{Q}_\tau, Q_\tau)\right]^p, \quad \tau = t, \ldots, T-1. \tag{22}$$

## A.2 MIXTURE GAUSSIAN DISTRIBUTION

This experiment is with the mixture Gaussian distribution, which imitates one step of the method (9). This simple example, working with a 2-dimensional and 1-time-stage Gaussian distribution, demonstrates the advantage of the subgradient method over traditional state-of-the-art mixed-integer solvers such as Gurobi. The marginal distribution $\widetilde{\lambda}_t$ is supported on five points $z^s$, and the conditional distributions $Q_t(\cdot|z^s)$, $s = 1, \ldots, 5$, are normal with the parameters:

$$\mu_1 = \begin{bmatrix} 0 \\ 0 \end{bmatrix}, \quad \mu_2 = \begin{bmatrix} 4 \\ -1 \end{bmatrix}, \quad \mu_3 = \begin{bmatrix} -3 \\ 3 \end{bmatrix}, \quad \mu_4 = \begin{bmatrix} 2.5 \\ 2.5 \end{bmatrix}, \quad \mu_5 = \begin{bmatrix} -1 \\ -2 \end{bmatrix}.$$

$$\sigma_1 = \begin{bmatrix} 0.5 & -0.2 \\ -0.2 & 0.5 \end{bmatrix}, \sigma_2 = \begin{bmatrix} 2 & 0 \\ 0 & 2 \end{bmatrix}, \sigma_3 = \begin{bmatrix} 1 & -0.1 \\ -0.1 & 1 \end{bmatrix}, \sigma_4 = \begin{bmatrix} 2 & 0.5 \\ 0.5 & 2 \end{bmatrix}, \sigma_5 = \begin{bmatrix} 1.6 & -1.2 \\ -1.2 & 1.6 \end{bmatrix}.$$

We set $\alpha^{(0)} = 0.01$, $\varepsilon = 10^{-7}$, $\varkappa_1 = 0.35$, and $\varkappa_2 = 0.35$. The potential representative points $\{\zeta^k\}_{k=1,\ldots,K}$ were Sobol lattice points. For illustration, we use the lattice points that cover the entire graph, even if some are obviously not necessary. To find the optimal values of $\beta$ and $\gamma$ in problem (11), we used the mixed integer programming (MIP) solver Gurobi and Algorithm 1. In Figures 2–4, the subfigures (a) show the sample points $\{x^{si}\}$ in five colors corresponding to the five Gaussian distributions and the potential locations of the representative particles. The subfigures (b) and (c) display the sample points and the grid points $\{z^k\}$ (black dots) selected by the MIP solver and the subgradient method, respectively. Table 2 provides the total numbers of the variables $\beta$ and $\gamma$, the solution times of both methods (in seconds), and the values of the Wasserstein distance $W_1$ of the solutions obtained to the colored cloud of particles. As the number of variables increases, the MIP solver takes an increasingly long time and becomes inapplicable. We further evaluated the effectiveness of the subgradient method on the multivariate Gaussian distribution and reported the results in Table 3, including the distribution's dimension. We also provide duality gap estimates, obtained as sketched below (14).

All numerical results were obtained using Python (Version 3.7) on a Macintosh HD laptop with a 2.9 GHz CPU and 16GB memory. In none of the experiments, the *stochastic* subgradient method (sketched on p. 6) was competitive.

Table 2: Comparison of the MIP solver and the subgradient method

| dim($\beta$) | dim($\gamma$) | MIP (s) | subgradient (s) | MIP $W_1$ | subgradient $W_1$ |
|---|---|---|---|---|---|
| 128000 | 256 | 5.17 | 0.96 | 0.654 | 0.644 |
| 512000 | 512 | 60.31 | 18.09 | 0.470 | 0.485 |
| 5120000 | 2048 | 556.26 | 346.57 | 0.246 | 0.272 |
| 20480000 | 4096 | - | 5130.23 | - | 0.222 |

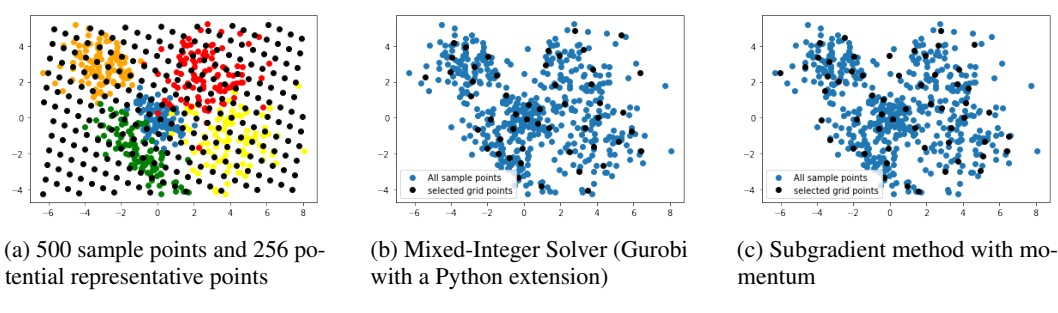

(a) 500 sample points and 256 potential representative points

(b) Mixed-Integer Solver (Gurobi with a Python extension)

(c) Subgradient method with momentum

Figure 2: $\dim(\beta) = 128000$, $\dim(\gamma) = 256$, and 51 selected particles.

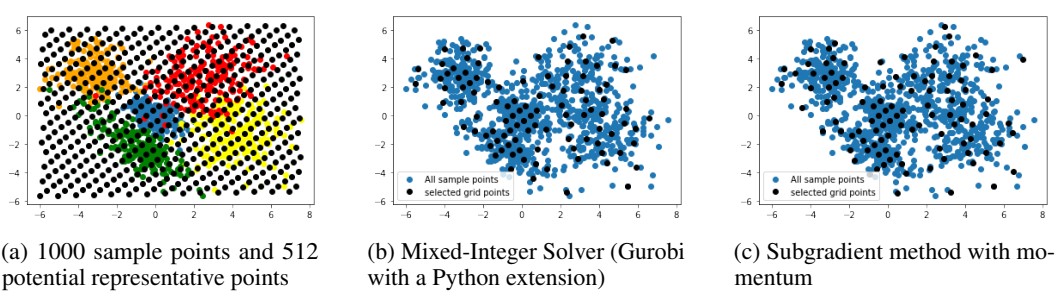

(a) 1000 sample points and 512 potential representative points

(b) Mixed-Integer Solver (Gurobi with a Python extension)

(c) Subgradient method with momentum

Figure 3: $\dim(\beta) = 512000$, $\dim(\gamma) = 512$, and 102 selected particles

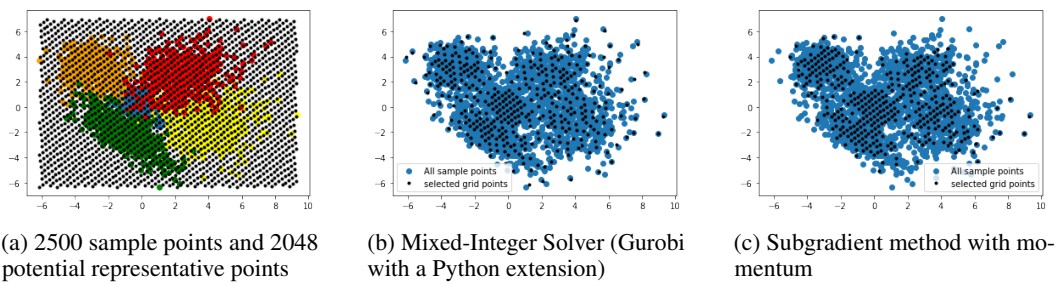

(a) 2500 sample points and 2048 potential representative points

(b) Mixed-Integer Solver (Gurobi with a Python extension)

(c) Subgradient method with momentum

Figure 4: $\dim(\beta) = 5120000$, $\dim(\gamma) = 2048$, and 409 selected particles

Table 3: Grid point selection with the subgradient method on multivariate Gaussian distribution

| dim | $\dim(\beta)$ | $\dim(\gamma)$ | subgradient (s) | subgradient $W_1$ | duality gap |
|---|---|---|---|---|---|
| 3 | 20000000 | 4000 | 2661 | 0.361 | 0.01208 |
| 4 | 31500000 | 4500 | 19493 | 0.564 | 0.00109 |
| 5 | 40000000 | 5000 | 11922 | 0.830 | 0.00388 |

