# OpenReview forum: "Fast Stochastic Kernel Approximation by Dual Wasserstein Distance Method"
_ICLR.cc/2024/Conference — ICLR 2024 Conference Withdrawn Submission_

### Official Review · Reviewer_Vxjx · 2023-10-28

**Soundness:** 4 excellent
**Presentation:** 3 good
**Contribution:** 3 good
**Rating:** 8
**Confidence:** 3

**Summary:**

This paper has a few contributions. First the authors introduce a way to measure distances between kernels via the Wasserstein distance. This distance is then leveraged in a sampling scheme of a Markov process.

**Strengths:**

The authors do a good job of incrementally introducing the complexity of the problem. Some of the problem statements are quite dense but they are motivated and presented well. The paper thus tells a full story with reasonable conclusions.

The distance for kernels is fairly novel (although it really is just an expected Wasserstein distance) but the authors treat it as a tool without examining its efficacy. Perhaps this is a tangent but demonstrating properties of this distance would have been great. This distance is a strength though because of the strength inherent to Wasserstein distance. It could perhaps be well applied in future works.


Minor comment: the conclusions start with a typo "W"

**Weaknesses:**

The notation can be hard to follow due to all the indices and characters, admittedly this could be unavoidable though. One thing which is avoidable though is switching to using "j" as iterations.

For definition 2.1 it would be beneficial to the authors to make it clear this is a contribution

I understand this a limited space issue but i wish the experimental results were more involved. I do see there is more results in the appendix, though. The experimental results are a bit underwhelming given the very nice theory presented earlier.

**Questions:**

In (11a) d_{sik} is defined as the distance between x and zeta, I am unclear as to why it's zeta and not z, any elaboration would be appreciated.

Since the authors propose solving their optimization problem via the dual rather than the primal, my question would normally involve the gap; the authors claim in 3.1 that for large dimensions the gap is minimal, could the authors elaborate on this?

---

> ### Author Response · Authors · 2023-11-17
> **Answers to Questions and Clarification**
>
> Thank you for your insightful review. Here are our answers to your questions:
> 1. The distances $d_{sik}$ are indeed between $x^{si}$ and the potential representative point $\zeta^{k}$. Only after $\gamma^k$ is found to be 1, the point $\zeta^k$ will become one of the $z^j$'s.
> 2. Our findings indicate that the gap is negligible for most examples we tested. For your information, the upper bound of the duality gap is recorded in Table 3 as well. In the worst-case scenario, even if the gap is substantial, our approach outperforms random particle selection, as seen in particle filter algorithms. One point that requires clarification is our assertion that "for large dimensions, the gap is minimal, as our experience demonstrates." Here, we specifically refer to the dimensions of primal variables $\gamma$ and $\beta$ and dual variables $\theta$. It's important to note that the dimensions of particles x and z do not directly impact the mixed-integer programming problem. In this context, the distance matrix between x and z contains the coefficients we need for the computation. This rationale underlines why we consider a numerical example on a Gaussian mixture distribution more appropriate for comparing the performances of the subgradient method and the integer programming solver. In this example, we have the flexibility to increase the dimensions of primal and dual variables, and the selected particles by both methods can be easily visualized.

---

> > ### Comment · Reviewer_Vxjx · 2023-11-20
> >
> > Thank you for the clarifications.

---

### Official Review · Reviewer_B8LX · 2023-11-02

**Soundness:** 1 poor
**Presentation:** 1 poor
**Contribution:** 1 poor
**Rating:** 1
**Confidence:** 5

**Summary:**

This paper proposed a generalized Wasserstein distance to quantify the difference between transition kernels of Markov system. The authors also propose an optimization algorithm that approximates system true dynamics efficiently. The effectiveness of this approach is validated in a simple numerical example.

**Strengths:**

- The problem itself seems to be an interesting topic. The modeling formulation seems sound. The problem has clear applications, such as optimal stopping problem as studied in Section 4.

**Weaknesses:**

I am regret to tell the authors that I must recommend rejection of this paper. The overall writing of this paper  remains to be improved. Besides, for the theoretical part,
- The authors solve the mixed-integer linear optimization formulation (11) by relaxing the boolean variable constraint $\gamma\in\\{0,1\\}$ with $\gamma\in[0,1]$. However, such a relaxation will usually induce large optimization error. Besides, the authors claim that "However, for large dimensions, the gap is minimal, as our experience demonstrates." When I look at the numerical results in Table 2, the optimization gap actually increases when the data dimension increases. Therefore, I think Section 3 does not present a reasonable approach for optimization.
- Several citations are missing. For example, in the last paragraph of Section 2, the authors introduced particle selection problem using the Wasserstein distance but did not include a citation. It is confusing to tell the specific application or problem formulation the authors are referring to.
- Numerical study is too stylized.

**Questions:**

N/A

---

> ### Author Response · Authors · 2023-11-17
> **No Response**
>
> This review does not merit a response.

---

> > ### Author Response · Authors · 2023-11-23
> > **Clarification**
> >
> > As the Area Chair prompted us, we are responding to the criticism of B8LX.
> > Indeed, our approach does not solve the mixed-integer optimization problem (11) exactly, because our dual approach solves its linear programming relaxation. This is our main idea. Its advantage is that the dual subgradient method of section 3 scales very well; it does not even involve matrix-vector multiplication. We openly admit a minor negative: relaxation is solved. However, the number of representative points $M_{t+1}$ in (11e) is not a hard modeling parameter; if we use 1003 points instead of 1000, the model is equally good. This, after the dual method converges, we can see how many $\gamma$'s were set to 1, and increase $M_{t+1}$ accordingly. Then, with the new $M_{t+1}$, the gap is zero and our solution from Algorithm 1 is optimal. But to satisfy the curiosity of the Reviewer, we estimated in Table 3 the gap in the following way. We reset to 0 the "surplus" $\gamma$'s for which the expression in braces {...} in (13) is closest to 0, and we reassigned other variables as described below (13). This provides a feasible integer solution and an upper bound. The gap is minimal. Why? Because of very many points, the probability mass reassigned is small, and the neighboring points are almost as good as the deleted "surplus" points.

---

> > > ### Comment · Reviewer_B8LX · 2023-11-23
> > > **Comments after rebuttal**
> > >
> > > I have read the authors' rebuttal and revised manuscript. Unfortunately, the replies do not change my opinion and the revised manuscript is still hard to read. So I decide to keep my score.

---

### Official Review · Reviewer_uLtp · 2023-11-10

**Soundness:** 2 fair
**Presentation:** 1 poor
**Contribution:** 2 fair
**Rating:** 5
**Confidence:** 4

**Summary:**

This paper proposes a new kernel based on Wasserstein distance for Markov systems. More specifically, the author proposes a Wasserstein distance between two kernels $Q$ and $\tilde{Q}$ (definition 2.1). Then, the computationally efficient algorithm is provided and the efficacy of the proposed method is demonstrated in numerical problems.

The approach seems to be new. However, it has several problems needed to be addressed further. Moreover, the experimental section can be further expanded

**Strengths:**

1. The problem in this paper is interesting.

**Weaknesses:**

1. The paper is generally poorly written and contains numerous typos in both the explanations and equations.
2. The experimental section could benefit from additional elaboration and expansion.

**Questions:**

1. The definition 2.1 is very confusing. More specifically, it explains $Q(\dot | x)$ is a kernel that transforms $x$ to the probability measure. It seems $Q(\dot | x)$ is a probability density function estimated and it does not look like a kernel functions. Could you please elaborate this part?
2. Equation (9) presents a variation of the Wasserstein barycenter problem. However, the manuscript lacks a comprehensive discussion of the Wasserstein barycenter, and it would be beneficial to explicitly address this concept. Additionally, a comparison of the proposed method with relevant baselines is warranted.
3. Numerous typographical errors are present in the manuscript. For instance, on page 1, there is a repeated "the the." Equation (6) is lacking the variable $p," and these issues need prompt correction for the manuscript to maintain clarity and accuracy.

---

> ### Author Response · Authors · 2023-11-17
> **Answers to the Questions**
>
> 1. In probability theory, a stochastic kernel, also referred to as a Markov kernel or probability kernel, functions as a map that assumes a role akin to that of a transition matrix in the theory of finite-state Markov chains. The formal definition is:
> Let $(X, \mathcal{A})$ and $(Y, \mathcal{B})$ be measurable spaces. A Markov kernel with source $(X, \mathcal{A})$ and target $(Y, \mathcal{B})$ is a map $Q: \mathcal{B} \times X \rightarrow[0,1]$ with the following properties:
> (a) For every event $B \in \mathcal{B}$, the map $x \mapsto Q(B|x)$ is $\mathcal{A}$-measurable;
> (b) For every point $x \in X$, the map $B \mapsto Q(B| x)$ is a probability measure on $(Y, \mathcal{B})$.
> Some Markov kernels are indeed defined by kernel functions $k(x,y)$, such that $Q(dy|x) = k(x,y) \mu(dy)$ with a base measure $\mu$ on $Y$. The misunderstanding may have originated from this special case. We do not work with kernel functions.
>
> 2. Equation (9) does not represent a variation of the Wasserstein barycenter problem. While our research may bear a superficial resemblance to the Wasserstein barycenter problem, our focus differs dramatically. Wasserstein barycenters offer a geometrically meaningful approach to aggregating probability distributions, grounded in optimal transport theory. However, our methodology diverges by not aiming to identify an aggregated measure that best captures multiple distributions, but rather a simplified kernel whose values approximate the values of the original kernel.  As demonstrated by Theorem 2.2 and our numerical example, our approach is directly applicable to Markov systems with multiple time stages, a scenario where the Wasserstein barycenter problem falls short. Although we cite papers related to the Wasserstein barycenter problem, they are only tangentially related to our goals.
>
> 3. The two typos have been corrected. Thank you for bringing them to our attention. Regarding Equation (6), we maintain that the variable "p" was not missing; it features in the definition of the matrix $D$.

---

> > ### Comment · Reviewer_uLtp · 2023-11-22
> > **Thank you for the clarification**
> >
> > Thank you for your clarification. 1 and 3 are cleared. I missed the p notation in the definition of d_{ij}.
> >
> > For the response 2, I agree that the original problem is not from the barycenter problem. However, it results in a barycenter problem. Thus, I wonder how the existing barycenter method performs your problem. This is not clear yet. Even though it is superficially related to barycenter, it would be great to discuss and compare with the barycenter problem. This will make a great benefit to machine learning community.
> >
> > Since the author clarified some misunderstandings, I would like to raise the score to 5.

---

> > > ### Comment · Reviewer_uLtp · 2023-11-23
> > > **Clarification of the paper**
> > >
> > > I re-read the method, and the method is not a barycenter problem. The problem looks similar, but not a barycenter problem.
> > >
> > > Unfortunately, the problem formulation part is quite confusing to understand the method and the experimental section can be further improved as I mentioned in the original comment. Thus, I will keep the current score.

---

> > > > ### Author Response · Authors · 2023-11-23
> > > > **Thank you for the response.**
> > > >
> > > > We are grateful for the reviewer’s insightful comments and suggestions.
> > > >
> > > > Our study introduces a new set of distance metrics based on the Wasserstein distance, aiming to present and explore this concept (section 2.2). We have also developed a subgradient-based algorithm for finding near-optimal solutions, which has shown excellent performance in our tests.
> > > >
> > > > While we have strived to explain the kernel distance clearly and rigorously, we appreciate that new concepts can be challenging to grasp without previous experience. Introducing novel ideas inherently involves some risk, but it is a necessary step for progress in our field.
> > > >
> > > > In conclusion, we thank the reviewer for your valuable input and emphasize the importance of acknowledging innovative contributions in our field and the broader machine learning community.

---

### Meta-Review · Area_Chair_eyW6 · 2023-11-30

**Metareview:**

This paper introduces a generalized Wasserstein distance to quantify dissimilarities between transition kernels in Markov systems. The authors also present an efficient optimization algorithm for approximating true system dynamics. The proposed method is validated through a numerical example, demonstrating its effectiveness. A key contribution lies in the introduction of a method to measure kernel distances using Wasserstein distance, which is subsequently applied in a sampling scheme for Markov processes.

While the method shows promise, there is room for improvement in both the presentation and experimental aspects. A substantial revision of the paper is recommended to enhance its overall quality. I strongly encourage the authors to address these concerns, revising the manuscript thoroughly before resubmitting it to a future venue.

**Justification For Why Not Higher Score:**

N/A

**Justification For Why Not Lower Score:**

The proposed method demonstrates merit, introducing a novel formulation. Nevertheless, there is room for improvement in both the presentation and experimental sections. Therefore, acceptance of the paper in its current form is challenging.

---

### Decision · Program_Chairs · 2024-01-16

Reject